# Store-Operated Calcium Entry in Breast Cancer Cells Is Insensitive to Orai1 and STIM1 N-Linked Glycosylation

**DOI:** 10.3390/cancers15010203

**Published:** 2022-12-29

**Authors:** Jose Sanchez-Collado, Joel Nieto-Felipe, Isaac Jardin, Rajesh Bhardwaj, Alejandro Berna-Erro, Gines M. Salido, Tarik Smani, Matthias A Hediger, Jose J. Lopez, Juan A. Rosado

**Affiliations:** 1Department of Physiology, Institute of Molecular Pathology Biomarkers, University of Extremadura, 10003 Caceres, Spain; 2Membrane Transport Discovery Lab, Department of Nephrology and Hypertension and Department of Biomedical Research, University of Bern, CH-3010 Bern, Switzerland; 3Department of Medical Physiology and Biophysic, Institute of Biomedicine of Sevilla, 41013 Sevilla, Spain

**Keywords:** Orai1, STIM1, glycosylation, Ca^2+^ entry, breast cancer cells, posttranslational modifications, store-operated Ca^2+^ entry

## Abstract

**Simple Summary:**

Breast cancer cells exhibit several differences in store-operated Ca^2+^ entry (SOCE) as compared to non-tumoral breast epithelial cells due to altered expression and post-translational modification of STIM proteins and Orai channels, as well as their modulators. The aim of this study was to analyze Orai1 and STIM1 N-linked glycosylation in SOCE in breast cancer cells and to ascertain the potential functional relevance of this post-translational modification in the development of cancer hallmarks. Using glycosylation-deficient STIM1 and Orai1 mutants we have found SOCE in breast cancer cells is insensitive to N-linked glycosylation of these proteins, a mechanism that might be relevant to evade apoptosis.

**Abstract:**

N-linked glycosylation is a post-translational modification that affects protein function, structure, and interaction with other proteins. The store-operated Ca^2+^ entry (SOCE) core proteins, Orai1 and STIM1, exhibit N-glycosylation consensus motifs. Abnormal SOCE has been associated to a number of disorders, including cancer, and alterations in Orai1 glycosylation have been related to cancer invasiveness and metastasis. Here we show that treatment of non-tumoral breast epithelial cells with tunicamycin attenuates SOCE. Meanwhile, tunicamycin was without effect on SOCE in luminal MCF7 and triple negative breast cancer (TNBC) MDA-MB-231 cells. Ca^2+^ imaging experiments revealed that expression of the glycosylation-deficient Orai1 mutant (Orai1N223A) did not alter SOCE in MCF10A, MCF7 and MDA-MB-231 cells. However, expression of the non-glycosylable STIM1 mutant (STIM1N131/171Q) significantly attenuated SOCE in MCF10A cells but was without effect in SOCE in MCF7 and MDA-MB-231 cells. In non-tumoral cells impairment of STIM1 N-linked glycosylation attenuated thapsigargin (TG)-induced caspase-3 activation while in breast cancer cells, which exhibit a smaller caspase-3 activity in response to TG, expression of the non-glycosylable STIM1 mutant (STIM1N131/171Q) was without effect on TG-evoked caspase-3 activation. Summarizing, STIM1 N-linked glycosylation is essential for full SOCE activation in non-tumoral breast epithelial cells; by contrast, SOCE in breast cancer MCF7 and MDA-MB-231 cells is insensitive to Orai1 and STIM1 N-linked glycosylation, and this event might participate in the development of apoptosis resistance.

## 1. Introduction

Among the mechanisms involved in the modulation of cell function by physiological agonist, store-operated Ca^2+^ entry (SOCE) plays an essential role. SOCE is a mechanism for Ca^2+^ influx regulated by the filling state of the intracellular Ca^2+^ stores. Intracellular Ca^2+^ store depletion leads to the activation of the two core proteins involved in the SOCE machinery: (1) the stromal interaction molecule-1 (STIM1), located in the membrane of the endoplasmic reticulum (ER), sensing intraluminal ER Ca^2+^ levels, that communicates the filling state of the Ca^2+^ stores to (2) Orai1, the pore-forming subunit of the Ca^2+^ release-activated Ca^2+^ (CRAC) channels that conducts the highly Ca^2+^ selective current I_CRAC_ [1,2,3,4,5,6].

The regulation of Ca^2+^ influx through CRAC channels is essential to generate Ca^2+^ signals that match the strength of agonist stimulation and to maintain intracellular Ca^2+^ homeostasis. This modulation involves the participation of the mammalian homologs of Orai1, Orai2 and Orai3, and STIM1, STIM2 [7,8,9,10]. In addition, a variety of regulatory proteins are involved in the modulation of CRAC channels. Regulatory proteins include SARAF (SOCE-associated regulatory factor), which is involved in the mechanism underlying slow Ca^2+^-dependent inactivation of CRAC channels [11,12,13], CRACR2A (CRAC regulator 2A), STIMATE, septins, golli and ORMDL3 (Orosomucoid like 3) (for a review see [14]). Similarly, a variety of post-translational modifications, including phosphorylation at serine or tyrosine residues [15,16,17,18], S-glutathionylation [19], redox modulation and O- and N-linked glycosylation modulate CRAC channels (for a review see [20]). N-linked glycosylation consists in the covalent attachment of an oligosaccharide to the amide nitrogen of an Asn residue that mostly occurs in the ER and Golgi. Both Orai1 and STIM1, as well as STIM2, are susceptible to be N-linked glycosylated. STIM1 is N-linked glycosylated at N131 and N171, located within the EF-SAM domain, while Orai1 contains a single N-linked glycosylation site at N223. The functional role of Orai1 N-linked glycosylation is controversial and exhibits cell type-specificity. In HEK-293 cells, introduction of a glycosylation-deficient mutant of Orai1 (Orai1N223A) does not significantly alter Orai1 function or localization [21]. In contrast, in fibroblasts derived from SCID (severe combined immunodeficiency) patients, overexpression of the Orai1N223A mutant enhances SOCE as compared to Orai1 wild type [22] and similar results were found in Jurkat T and CHO cells [23]. The reason of this apparent discrepancy is unclear, but it has been attributed to differences in the sialic acid-binding partners expressed [23]. On the other hand, STIM1 N-linked glycosylation has been reported to facilitate EF-SAM destabilization and oligomerization through structural changes in the core α8 helix, in the EF-SAM domain, resulting in enhanced Ca^2+^ influx through CRAC channels [24,25]. Abnormal N-linked glycosylation of STIM1 and Orai1 has been observed in a variety of pathologies, including myopathies with tubular aggregates or septic myocardial depression [26,27].

Cancer cells are mostly characterized by uncontrolled proliferation, invasion of local tissue and spreading to other organic territories [28]. The mechanisms underlying these changes involve abnormal Ca^2+^ homeostasis mediated by up- or downregulation of Ca^2+^ channels and regulatory proteins, including the SOCE core proteins STIM1 and Orai1. Breast cancer is one of the most common cancer types in women. Orai1 is upregulated in estrogen receptor positive (ER+) and triple negative breast cancer cells [29,30] and this channel regulates the transcription of genes that contribute to cancer progression [31]. SOCE has been reported to be critical for the development of different breast cancer hallmarks, such as cell proliferation, migration and metastatic spread [29,30,32,33]. Aberrant SOCE has been associated to a variety of disorders, including cancer [34]. Furthermore, alterations in Orai1 glycosylation have been observed in cancer cells and overexpression of the enzyme β-galactoside α-2,6-sialyltransferase 1 (ST6GAL1) has been reported to play a relevant role in cancer invasiveness and metastasis. This enzyme glycosylates Orai1, which results in reduced SOCE [34].

In the present study we have investigated the role of STIM1 and Orai1 N-linked glycosylation in SOCE in non-tumoral breast epithelial cells as well as in ER+ and triple negative breast cancer (TNBC) cells. Here we show that while STIM1 and Orai1 N-linked glycosylation has a significant effect on SOCE in non-tumoral breast epithelial cells, both ER+ and triple negative breast cancer (TNBC) cells are unaffected by this mechanism. These findings provide evidence for a mechanism to evade the regulation of Ca^2+^ influx in breast cancer cells.

## 2. Materials and Methods 

### 2.1. Materials and Reagents 

Fura-2 acetoxymethyl ester (fura-2/AM) was purchased from Molecular Probes (Leiden, The Netherlands). Insulin, TG, rabbit anti-β-actin antibody (catalog number A2066, epitope: sequence 365–375 of human β-actin), epidermal growth factor, bovine serum albumin (BSA), rabbit polyclonal anti-Orai1 antibody (catalog number O8264, epitope: amino acids 288-301 of human Orai1), mouse monoclonal anti-PMCA antibody (clone 5F10; epitope: sequence 724–783 of human PMCA; catalog number MA3-914), HEPES, EGTA, caspase-3 substrate Z-DEVD-AFC and EDTA (ethylenedinitrilotetraacetic acid) were obtained from Sigma (Madrid, Spain). Polyclonal anti-caspse-3 (catalog number 9662) and anti-cleaved caspase-3 (Asp175) (catalog number 9664) antibodies were from Cell Signaling (Leiden, The Netherlands). Trypsin, penicillin/streptomycin, EZ-Link™ Sulfo-NHS-LC-Biotin, BCA protein assay kit and high-glucose Dulbecco’s modified Eagle’s medium (DMEM) were purchased from ThermoFisher Scientific (Waltham, MA, USA). Mouse monoclonal Anti-GOK/Stim1 antibody (Clone 44/GOK; catalog number 610954, epitope: amino acids: 25–139 of human STIM1) was purchased from BD Biosciences (San Jose, CA, USA). Tunicamycin was from Proquinorte (Zamudio, Spain). DharmaFECT was from Dharmacon (Lafayette, CO, USA). Horseradish peroxidase-conjugated goat anti-mouse immunoglobulin G (IgG) antibody and goat anti-rabbit IgG antibody were from Jackson laboratories (West Grove, PA, USA). Enhanced chemiluminescence detection reagents were from Pierce (Cheshire, UK). Bromodeoxyuridine (BrdU) cell proliferation assay kit was from BioVision. CMV-promoter STIM1-YFP was kindly provided by Christoph Romanin (University of Linz, Linz, Austria). CMV-promoter STIM1N131Q/STIM1N171Q mCherry plasmid was kindly provided by Peter Stathopulos (Western University, London, Ontario, Canada). MO70-Orai1 WT was kindly provided by Olivier Mignen (University of Brest, Brest, France). MO70 Orai1 N223A was a gift from Anjana Rao (Addgene plasmid # 21663; http://n2t.net/addgene:21663, accessed on 1 December 2022; RRID:Addgene_21663). All other reagents were of analytical grade.

### 2.2. Cell Culture and Transfections

CRISPR-generated Orai1 single-knockout MCF-7 and MDA-MB-231 cells, STIM1 single-knockout MCF-7 and MDA-MB-231 cells and parental MCF-7 and MDA-MB-231 cells were kindly provided by Rajesh Bhardwaj (Hediger Membrane Transport Discovery Lab, University of Bern, Switzerland) and cultured at 37 ℃ with a 5% CO_2_ in high-glucose DMEM supplemented with 10% (*v*/*v*) fetal bovine serum and 100 U/mL penicillin and streptomycin, as described previously [30]. MCF10A cells were purchased from ATCC and cultured at 37 °C with a 5% CO_2_ in Dulbecco’s Modified Eagle Medium-F12, supplemented with 5% (*v*/*v*) horse serum, 0.5 μg/mL hydrocortisone, 20 ng/mL epidermal growth factor, 10 μg/mL insulin, and 100 ng/mL cholera toxin [35].

For transient transfections, cells were grown to 60 to 80% confluency and transfected with the indicated plasmids using DharmaFECT kb transfection reagent and were used 48 h after transfection. For Western blotting, cells (2 × 10^6^) were plated in 100-mm petri dish and cultured for 48 h, while, for Ca^2+^ imaging and confocal analysis cells (4 × 10^5^) were seeded in a 35 mm six-well multidish.

### 2.3. Determination of Cytosolic Free-Ca^2+^ Concentration

Cells were loaded with fura-2 by incubation with 2 µM fura 2/AM for 30 min at 37 °C. Ca^2+^ determination was performed using an epifluorescence inverted microscope (Nikon Eclipse Ti2, Amsterdam, The Netherlands) with image acquisition and analysis software (NIS-Elements Imaging Software v.5.02.00, Nikon, Japan). Cells were superfused with HEPES-buffered saline containing 125 mM NaCl, 5 mM KCl, 1 mM MgCl_2_, 5 mM glucose, 25 mM HEPES, and pH 7.4, supplemented with 0.1% (*w*/*v*) BSA. Samples were alternatively excited at 340/380 nm with light from a xenon lamp passed through a high-speed monochromator (Optoscan ELE 450, Cairn Research, Faversham, UK). Fluorescence emission was detected at 505 nm using a sCMOS camera (PCO Panda 4.2 (Excelitas PCO GmbH, Kelheim, Germany)) and recorded using NIS-Elements AR software (Nikon, Amsterdam, The Netherlands). Fluorescence ratio (F340/F380) was calculated pixel by pixel [36]. TG-evoked Ca^2+^ release and entry were measured as the integral for 3 and 2½ min, respectively, of the rise in the fura-2 fluorescence ratio after the addition of TG or Ca^2+^, respectively.

### 2.4. Western Blotting

Cell lysates were analyzed by 10% SDS-PAGE. Proteins were electrophoretically transferred onto nitrocellulose membranes for subsequent probing. Nitrocelulose membranes were incubated overnight with blocking buffer (containing: 10% (*w*/*v*) BSA in Tris-buffered saline and 0.1% Tween 20 (TBST)) to block residual protein binding sites. Detection of Orai1, STIM1, caspase-3, cleaved caspase-3 and β-actin was performed by incubation with anti-Orai1 or anti-STIM1 antibodies diluted 1:1000 in TBST for 1 h, with the anti-caspase-3 or anti-cleaved caspase-3 antibodies diluted 1:1000 in TBST for 2 h, or by incubation with the anti-β-actin antibody diluted 1:2000 in TBST for 1 h. Blots were washed with TBST and were incubated for 1 h with horseradish peroxidase-conjugated goat anti-rabbit IgG antibody or goat anti-mouse antibody diluted 1:10,000 in TBST. Blots were then exposed to enhanced chemiluminescence reagents for 5 min. The density of bands was measured using ChemiDoc MP Imaging System (Bio-Rad, Madrid, Spain) and Fiji-ImageJ software (NIH, Bethesda, MD, USA).

### 2.5. Biotinylation of Cell Surface Proteins

The labeling and isolation of cell surface proteins were performed by surface biotinylation assay, as described previously [37]. Cells were washed three times with phosphate-buffered saline (PBS, NaCl 137 mM, KCl 2.7 mM, KH_2_PO_4_ 1.5 mM, Na_2_HPO_4_·2H_2_O 8 mM, pH 8). Cells were then incubated at 4 °C for 1 h with biotynilation buffer consisting of PBS supplemented with 1 mg/mL EZ-Link sulfo-NHS-LC-biotin. The reaction was terminated by addition of Tris base (50 mM). Cells were disrupted using Nonidet P-40 buffer and sonicated, and lysates were centrifuged (16,000× *g* for 5 min at 4 °C). Protein concentration was measured using BCA assay. Samples were incubated for 2 h with 50 µL streptavidin beads at 4 °C and re-suspended in Laemmli buffer. The biotinylated and non-biotinylated fractions were separated in 8% SDS-PAGE and Orai1 and PMCA surface expression were detected by Western blotting using specific antibodies.

### 2.6. Caspase-3 Activity Assay

To determine caspase-3 activity, 2 × 10^6^ cells were lysed using ice-cold Nonidet P-40 buffer and sonicated. Cell lysates were mixed with Nonidet P-40 (supplemented with 5 mM DTT and 8.25 mM of caspase-3 substrate Z-DEVD-AFC). Samples were incubated at 37 °C for 2 h. as described previously [38]. Cleaving of the caspase-3 substrate was determined with a fluorescence spectrophotometer with excitation wavelength of 400 nm and emission at 505 nm. Caspase-3 activity was calculated from the cleavage of its fluorogenic substrate following the manufacturer’s instructions. 

### 2.7. CRISPR/Cas9 Mediated Generation of STIM1 and Orai1 KO MDA-MB-231 and MCF-7 Cells

STIM1 KO and Orai1 KO MDA-MB-231 cells were generated and validated in [39]. MCF-7 STIM1 KO cells were generated using earlier described methods for generating STIM1 knockout [40] and MCF-7 Orai1 KO cells were generated following the methodology described in [41].

### 2.8. Statistical Analysis

Statistical analysis was performed using the Kruskal-Wallis test combined with Dunn’s post hoc test (GraphPad Prism Windows 8, San Diego, CA, USA). To compare two groups, the Mann-Whitney U test (or Student’s *t* test for the analysis of Ca^2+^ determinations) was used. *p* values < 0.05 were considered statistically significant.

## 3. Results

### 3.1. Orai1 and STIM1 Glycosylation in Breast Cancer and Non-Tumoral Breast Epithelial Cells 

We have analyzed native STIM1 and Orai1 N-linked glycosylation in non-tumoral breast epithelial cells, ER+, HER2 and TNBC cells. The normal breast epithelial cell MCF10A, the ER+ breast cancer cell lines T47D and MCF7, the HER2 cell line SKBR3 and the TNBC cell lines BT20, MDA-MB-468, MDA-MB-231 and HS578T were treated with PNGase F, to remove N-linked oligosaccharides [42], or the vehicle and then subjected to Western blotting with specific anti-STIM1 or anti-Orai1 antibodies. STIM1 and Orai1 are N-glycosylated and, as shown in Figure 1, Western blot analysis of whole-cell lysates with specific anti-STIM1 antibody revealed a band with smaller size in samples treated with PNGase F, corresponding to non-glycosylated STIM1. Multiple bands corresponding to Orai1 glycosylated were observed in untreated cells (Figure 1). These multiple bands almostly disappear upon treatment, with PNGase F mostly leaving two bands corresponding to the two Orai1 forms identified in mammalian cells, the full length Orai1α, with a predicted size of 33 kDa, and a shorter form lacking 63 amino acids at the N-terminus, Orai1β of 23 kDa [43]. Our results indicate that all the cells investigated show a similar STIM1 glycosylation pattern and cell type-specific Orai1 glycosylation patterns, as we found that in all the cells investigated, the pattern of Orai1 immunoreactivity appeared with distinct molecular masses (Figure 1).

### 3.2. Effect of Tunicamycin in TG-Induced Ca^2+^ Release and Entry in Breast Cancer and Non-Tumoral Breast Epithelial Cells

To investigate the possible functional role of Orai1 and STIM proteins N-linked glycosylation in breast cancer cells, we treated cells with tunicamycin, a nucleoside antibiotic that inhibits the initial step of N-linked glycosylation by preventing the transfer of N-acetylglucosamine-1-phosphate from UDP-GlcNAc to dolichol-P [44]. Non-tumoral breast epithelial MCF10A cells, ER+ breast cancer MCF7 cells and MDA-MB-231 TNBC cells were incubated overnight with 2 µM tunicamycin or the vehicle. The MCF7 and MDA-MB-231 cell lines were chosen to perform the study as they are commonly used breast cancer cell lines for in vitro studies on ER+ and triple negative breast cancers [45]. To confirm the effect of tunicamycin in protein glycosylation we assessed the pattern of STIM1, STIM2 and Orai1 immunoreactivity by Western blotting. As depicted in Figure 2a, in mock-treated cells STIM1 and STIM2 appear predominantly as a single band at the predicted size and, upon treatment with tunicamycin the predicted band appear slightly faint and the non-glycosylated proteins run at a lightly lower molecular mass. Orai1 exhibit a band pattern above 33 kDa, the predicted size of the full-length of Orai1 [43] in all the cells investigated, and treatment with tunicamycin decreases the intensity of the bands corresponding to glycosylated Orai1 and evidenced an increase in the 23 kDa band, the predicted size of the short form of Orai1, Orai1β [37,42,43] (Figure 2a).

SOCE in MCF10A and MDA-MB-231 cells has been reported to be strongly dependent on STIM1 and Orai1, while in MCF7 cells is mostly regulated by STIM1 and Orai3 [29,30]. In contrast to STIM1 and Orai1, Orai3 has no N-glycosylation consensus motifs [23]; therefore, we have focused our studies in the functional role of STIM1 and Orai1 glycosylation in SOCE in the breast cancer cell types as compared to MCF10A breast epithelial cells. MCF10A, MCF7 and MDA-MB-231 cells were incubated with tunicamycin or the vehicle to analyze the effect of glycosylation on TG-evoked Ca^2+^ release and entry. Cell treatment with 1 µM TG, an inhibitor of the sarco/endoplasmic reticulum Ca^2+^-ATPase (SERCA), induced a transient increase in the fura-2 fluorescence ratio as a result of Ca^2+^ release from the intracellular Ca^2+^ stores. Subsequent addition of extracellular Ca^2+^ leads to a rise in the fura-2 fluorescence ratio that is indicative of Ca^2+^ influx. As shown in Figure 2b, in MCF10A cells tunicamycin significantly enhanced TG-induced Ca^2+^ release from the intracellular stores but, by contrast, attenuated SOCE by 30% (*p* < 0.001; Student *t*-test). As depicted in Figure 2c, in MCF7 cells tunicamycin results in a significant increase in TG-evoked Ca^2+^ release but it did not modify SOCE. Similarly, tunicamycin was unable to alter SOCE in the TNBC cell line MDA-MB-231 although inhibition of protein glycosylation significantly attenuated TG-induced Ca^2+^ release (Figure 2d). These findings indicate that a blockade of protein glycosylation by tunicamycin exhibits cell-specific effects on the ability of these cells to accumulate Ca^2+^ into the intracellular stores or the Ca^2+^ leakage rate from the stores. Interestingly, SOCE in non-tumoral breast epithelial MCF10A cells was sensitive to treatment with tunicamycin while in breast cancer cells this mechanism exhibits resistance to inhibition of protein glycosylation.

Since tunicamycin attenuates SOCE in MCF10A, we have further investigated whether tunicamycin might influence the plasma membrane expression of Orai1 in these cells. MCF10A cells were treated overnight with tunicamycin or the vehicle and surface expression of Orai1 was assessed by biotinylation. As depicted in Appendix A, our results indicate that the plasma membrane location of Orai1 was significantly reduced by tunicamycin, thus suggesting that the effect of tunicamycin on SOCE in MCF10A involves the attenuation of Orai1 plasma membrane expression. We also performed this analysis in breast cancer MDA-MB-231 cells where the surface expression of Orai1 was not significantly altered by treatment with tunicamycin, which supports the lack of effect of tunicamycin on SOCE in these cells (Appendix A). Tunicamycin has been reported to inhibit protein synthesis by disturbing ER homeostasis due to accumulation of misfolded proteins [46]. Hence, we have further explored the possible effect of tunicamycin on Orai1 and STIM1 protein expression in MCF10A cells and MDA-MB-231 breast cancer cells. As shown in Appendix A, treatment with tunicamycin significantly attenuates Orai1 and STIM1 expression in MCF10A cells. In the presence of cycloheximide (CHX), a well-known inhibitor of protein synthesis [47], tunicamycin was unable to further decrease Orai1 and STIM1 expression significantly (Appendix A), which suggests that tunicamycin attenuates Orai1 and STIM1 protein synthesis in MCF10A cells. Similarly, in MDA-MB-231 cells, tunicamycin slightly attenuate Orai1 and STIM1 expression in the absence and presence of CHX (Appendix A). It is worth to mention that both CHX and tunicamycin have a smaller effect in MDA-MB-231 cells than in MCF10A cells, thus suggesting that Orai1 and STIM1 protein degradation is slower in MDA-MB-231 breast cancer cells (Appendix A).

### 3.3. Characterization of the Functional Role of Orai1 and STIM1 N-Linked Glycosylation in TG-Induced Ca^2+^ Release and Entry in Breast Cancer and Non-Tumoral Breast Epithelial Cells

As mentioned above, studies on Orai1 N-linked glycosylation have reported cell-specific effects on CRAC channel function and SOCE [21,22,23]. Hence, we have further explored the specific functional role of Orai1 glycosylation in SOCE in non-tumoral breast epithelial cells and breast cancer cells. To address this issue, we have expressed the glycosylation-deficient Orai1 mutant, Orai1N223A, in Orai1KO-MCF7 and Orai1KO-MDA-MB-231 cells as well as in MCF10A cells, which exhibit a very low endogenous Orai1 expression [42], as see Figure 2a. Expression of Orai1WT in the three cell lines results in the typical pattern of glycosylated Orai1 bands; however, the Orai1N223A mutant leads to two clear bands corresponding to the two forms of Orai1, Orai1α, the long form of 33 kDa, and Orai1β, the short form of predicted 23 kDa [43] (Figure 3a). As shown in Figure 3b–d, expression of the Orai1N223A mutant in non-tumoral MCF10A cells did not significantly modify TG-induced Ca^2+^ release or entry, and similar results were observed upon expression of the glycosylation-deficient Orai1 mutant in Orai1-KO MCF7 and Orai1-KO MDA-MB-231 cells. Transfected cells were analyzed for Orai1 expression by confocal microscopy. Cells transfected with either Orai1WT or Orai1N223A confirmed fluorescence labeling at the cytosol and at/by the plasma membrane (Figure 3e). These findings indicate that Orai1 glycosylation does not play any significant role in SOCE or the ability of accumulate Ca^2+^ in the intracellular stores in breast cancer or breast epithelial cells.

We have further investigated the specific functional role of STIM1 glycosylation in SOCE in tumoral and non-tumoral breast cells. To address this issue, we have expressed the glycosylation-deficient STIM1 mutant, STIM1N131/171Q, in STIM1KO-MCF7 and STIM1KO-MDA-MB-231 cells as well as in MCF10A cells. As shown in Figure 4a, expression of STIM1-YFP results in a single band, while, the STIM1N131/171Q-mCherry mutant leads to a slightly smaller band corresponding to unglycosylated STIM1, as well as other faint bands of smaller size of unknown nature. In MCF10A cells, due to the limited expression of Orai1 (see Figure 1), STIM1 or the STIM1N131/171Q mutant were co-expressed with Orai1. Expression of the STIM1N131/171Q mutant in non-tumoral MCF10A cells significantly attenuated TG-induced Ca^2+^ entry without having any effect on TG-evoked Ca^2+^ release from the intracellular stores, thus suggesting that STIM1 glycosylation plays a positive role in SOCE but have no effect on the ability to accumulate Ca^2+^ into the intracellular stores (Figure 4b). By contrast, expression of the glycosylation-deficient STIM1 mutant in STIM1-KO MCF7 and STIM1-KO MDA-MB-231 cells did not significantly alter either TG-induced Ca^2+^ release or SOCE (Figure 4c,d). Transfected cells were analyzed for STIM1 expression by confocal microscopy. Cells transfected with either STIM1-YFP or STIM1N131/171Q-mCherry confirmed fluorescence labeling at the cytosol (Figure 4e). Interestingly, co-transfection of MCF10A with STIM1N131/171Q and Orai1N223A expression plasmids reduced SOCE to a similar extent as expression of STIM1N131/171Q and WT-Orai1 (Figure 5). These findings indicate that STIM1 glycosylation plays a relevant and predominant role in SOCE in non-tumoral breast epithelial cells, but breast cancer cells escape from that regulatory mechanism.

### 3.4. Role of STIM1 Glycosylation in TG-Evoked Apoptosis in Breast Cancer and Non-Tumoral Breast Epithelial Cells

Next, we have explored the possible role of STIM1 N-linked glycosylation in TG-induced apoptosis in MCF10A cells and the breast cancer cell lines MCF7 and MDA-MB-231. Apoptosis was determined by estimation of caspase-3 activity, an important mediator of apoptosis [48,49]. In MCF10A cells, STIM1 or the glycosylation-deficient STIM1 mutant was co-expressed with Orai1 to maintain the stoichiometry. Our results indicate that treatment of MCF10A cells expressing STIM1-YFP for 24 h with TG results in a significant increase in caspase-3 activity (Figure 6). Caspase-3 activity in resting MCF10A cells expressing the STIM1N131/171Q mutant was similar to that observed in cells expressing STIM1-YFP; however, the ability of TG to induce caspase-3 activation in cells expressing the glycosylation-deficient STIM1 mutant was significantly attenuated (Figure 6). By contrast, TG-evoked caspase-3 activation was not found to be significantly different in MCF7 and MDA-MB-231 cells expressing either STIM1-YFP or the glycosylation-deficient STIM1N131/171Q mutant (Figure 6). Similar results were observed when we analyzed caspase-3 cleavage (Figure 7). Interestingly, the ability of TG to induce apoptosis in breast cancer cells expressing STIM1-YFP was significantly smaller than that observed in non-tumoral cells (Figure 6 and Figure 7). These findings indicate that impairment of STIM1 glycosylation attenuates the activation of caspase-3 in non-tumoral cells, while in breast cancer cells, TG-evoked caspase-3 activation, which was significantly smaller than in non-tumoral cells, was unaffected by impairment of STIM1 N-linked glycosylation.

## 4. Discussion

N-linked glycosylation involves the attachment of an olygosaccharide to the amide nitrogen of an asparagine residue by enzymatic reactions and plays a relevant role in protein structure and function. The type of the oligosaccharide attached is protein- and cell type-specific [50]. STIM1 and Orai1, the core SOCE proteins, are glycosylated proteins. Orai1 exhibits a single glycosylation site at N223 while STIM1 contains three consensus sites for N-linked glycosylation at N131, N171 and N658; however, only N131 and N171 can be glycosylated in vivo [22,51]. Aberrant STIM1 and Orai1 glycosylation have been reported in a number of disorders, for instance, it has been detected in myopathies with tubular aggregates and in sepsis-induced myocardial depression [26,27]. Furthermore, abnormal glycosylation has been associated to cancer progression and metastasis. In breast cancer cells, inhibitors of α-glucosidases have been reported to attenuate STIM1 expression, which significantly reduces SOCE and cell migration [52].

The functional role of Orai1 N-linked glycosylation in SOCE is cell type-specific, with reported inhibitory role or no function at all depending on the cell type investigated [19,20,21]. Concerning STIM1 N-linked glycosylation, studies using a variety of glycosylation-deficient STIM1 mutants have revealed that STIM1 glycosylation plays a positive role in SOCE, except for the STIM1N131D/N171Q mutant that induces a gain of function in patch-clamp experiments but not in Ca^2+^ imaging experiments [21]. The functional role of STIM1 glycosylation in SOCE has been attributed to structural perturbations within the EF-SAM core resulting in enhanced SOCE [22,23].

SOCE plays essential functional roles in breast cancer cell biology and the development of different breast cancer hallmarks, including cell viability and survival, migration, proliferation and apoptosis resistance [36,52,53,54]; however, the possible cellular effects of Orai1 and STIM1 N-linked glycosylation in breast epithelial and breast cancer cells remains unknown. Here we report that expression of a glycosylation-deficient Orai1 mutant (Orai1N223A) in Orai1KO luminal breast cancer MCF7 cells and TNBC MDA-MB-231 cells, as well as in non-tumoral breast epithelial MCF10A cells, is not essential for the activation of SOCE. These findings are consistent with the observations in HEK-293 cells [21,22]. As previously reported in HEK-293 cells [21,24], expression of the non-glycosylable STIM1N131/171Q mutant in non-tumoral MCF10A cells significantly reduces SOCE, which strongly supports a positive role for STIM1 glycosylation in SOCE in these cells. Interestingly, expression of the glycosylation incompetent STIM1N131/171Q mutant in STIM1KO MCF7 cells and STIM1KO MDA-MB-231 cells was without effect on SOCE. To our knowledge, this is the first description of a lack of effect of STIM1 glycosylation in SOCE in a given cell type. The results observed in cells expressing the STIM1N131/171Q mutant are consistent with the effect of tunicamycin in non-tumoral and breast cancer cells. In addition, these findings might explain the effect of tunicamycin in SOCE in MCF10A cells. Alternatively, we have found that tunicamycin attenuates STIM1 expression in MCF10A cells, as well as Orai1 plasma membrane expression, probably associated to a reduction in Orai1 synthesis. Consistent with the previously reported cell type-specific effect of N-glycosylation in SOCE, tunicamycin has been reported to enhance SOCE in a variety of cells, including the VSC4.1 (motoneuron-neuroblastoma hybrid) cell line [55] and lymphoidal cells [56].

In breast cancer cells, Orai1 and STIM1 proteins commonly appear as glycosylated proteins, as demonstrated by the electrophoretical mobility of these proteins in untreated cells, and the insensitivity of breast cancer cells to STIM1 glycosylation might reflect a functional adaptation of breast tumoral cells. Here we show that impairment of STIM1 N-linked glycosylation protect non-tumoral breast epithelial cells from the development of apoptosis upon treatment with TG, a finding that is consistent with the attenuation of TG-evoked Ca^2+^ entry in these cells [29,30]. However, in breast cancer cells the ability of TG to induce apoptosis is significantly smaller than in non-tumoral cells, revealing a certain degree of apoptosis resistance. Under these conditions, impairment of STIM1 glycosylation was without effect on TG-induced apoptosis, in the same way as it failed to attenuate TG-evoked Ca^2+^ influx. Summarizing, our results indicate that STIM1 N-linked glycosylation plays a relevant role in the activation of TG-evoked SOCE in non-tumoral breast epithelial cells but, by contrast, STIM1 glycosylation is without effect in the regulation of Ca^2+^ influx in breast cancer cells, an adaptative mechanism that might play an important role in apoptosis resistance in breast tumoral cells, as the lack of sensitivity of SOCE to STIM1 glycosylation in breast cancer cells deprives the apoptotic event of a regulatory mechanism. 

## 5. Conclusions

Our results reveal that in non-tumoral breast epithelial cells STIM1 N-linked glycosylation plays an essential role for full SOCE activation. By contrast, breast cancer cells scape from this regulatory mechanism, so that, SOCE is independent on STIM1 or Orai1 N-linked glycosylation. As STIM1 glycosylation plays a relevant role in the development of apoptosis upon treatment with the SERCA inhibitor TG in non-tumoral breast cells, the lack of effect of STIM1 glycosylation in SOCE and apoptosis in breast cancer cells might be attributed to a mechanism to evade apoptosis in these cells.

## Figures and Tables

**Figure 1 cancers-15-00203-f001:**
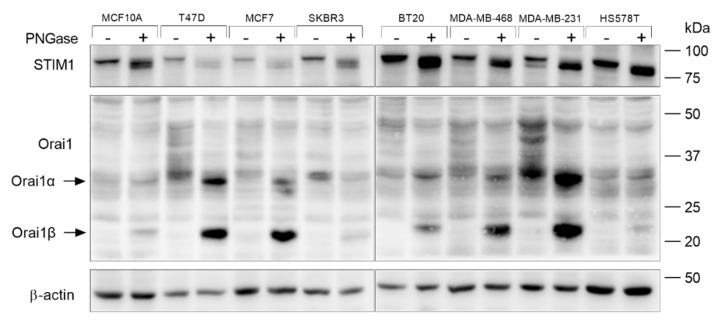
Characterization of the Orai1 and STIM1 N-linked glycosylation pattern in non-tumoral breast epithelial cells and breast cancer cells. MCF10A, T47D, MCF7, SKBR3, BT20, MDA-MB-468, MDA-MB-231 and HS578T cells were lysed. Cell lysates treated in the absence or presence of PNGase F, as indicated, and analyzed by Western blotting using specific anti-STIM1 or anti-Orai1 antibody. The membrane was reprobed with anti-β-actin antibody for protein loading control. Molecular masses indicated on the right were determined using molecular-mass markers run in the same gel.

**Figure 2 cancers-15-00203-f002:**
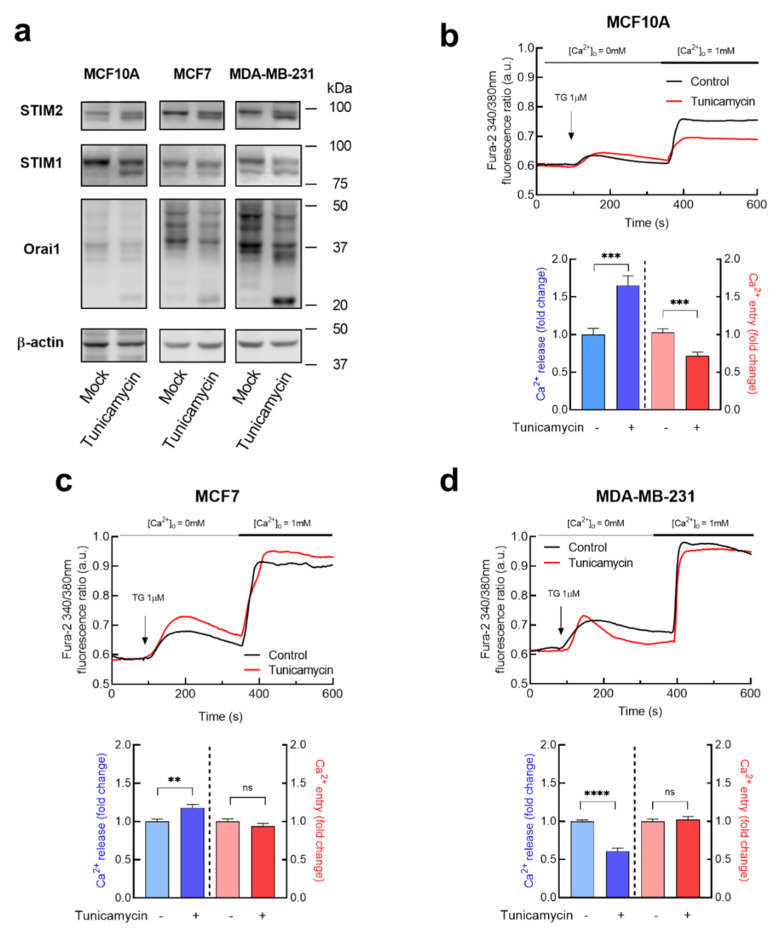
Effect of tunicamycin on Ca^2+^ release and entry in non-tumoral breast epithelial cells and breast cancer cells. (**a**) MCF10A, MCF7 and MDA-MB-231 cells were incubated with 2 µM tunicamycin or the vehicle (Mock) overnight, as indicated, and then were lysed. Cell lysates were analyzed by Western blotting using specific anti-STIM2 antibody, anti-STIM1 antibody or anti-Orai1 antibody. The membrane was reprobed with anti-β-actin antibody for protein loading control. Molecular masses indicated on the right were determined using molecular-mass markers run in the same gel. (**b**–**d**) MCF10A (**b**), MCF7 (**c**) and MDA-MB-231 (**d**) cells were incubated overnight with 2 µM tunicamycin or the vehicle (Mock), as indicated. Fura-2-loaded cells were perfused with a Ca^2+^-free medium (100 µM EGTA was added) and stimulated with TG (1 µM). Six minutes later Ca^2+^ (1 mM) was added to the extracellular medium to initiate Ca^2+^ entry. Bar graphs represent TG-stimulated Ca^2+^ release and entry in MCF10A (**b**), MCF7 (**c**) and MDA-MB-231 (**d**), expressed as fold change experimental over control. Data are mean ± SEM of 40 cells/day/3–5 days and were statistically analyzed using Student *t*-test. ** *p* < 0.01, *** *p* < 0.001 and **** *p* < 0.0001 compared to Ca^2+^ release or entry in control cells.

**Figure 3 cancers-15-00203-f003:**
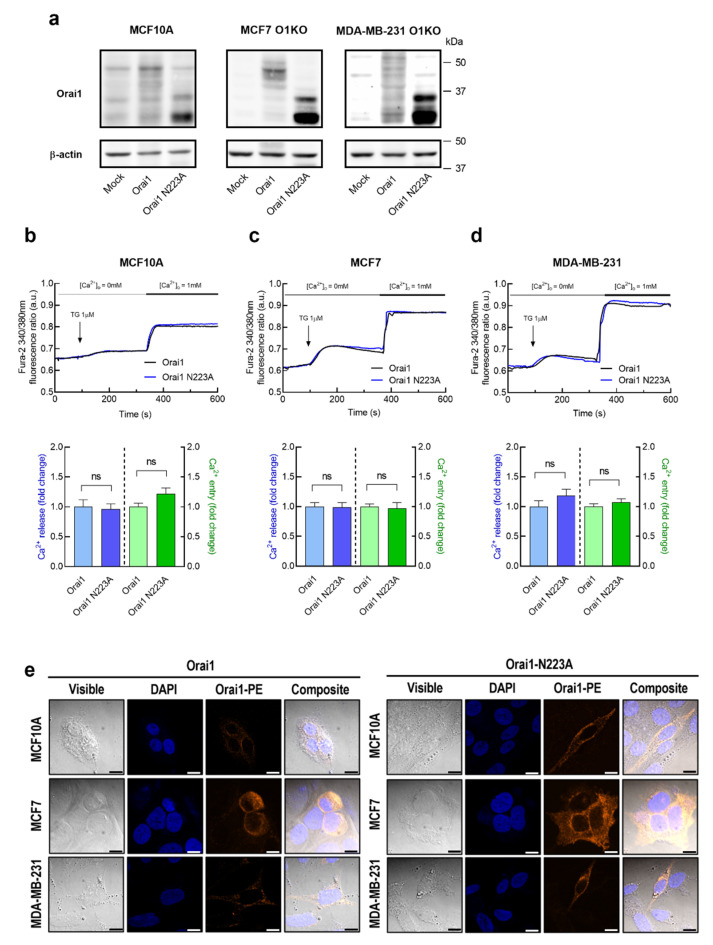
Characterization of Ca^2+^ mobilization in non-tumoral breast epithelial and breast cancer cells expressing Orai1WT and Orai1N223A. (**a**) MCF10A, MCF7-Orai1KO (MCF7 O1KO) and MDA-MB-231-Orai1KO (MDA-MB-231 O1KO) cells were transfected with Orai1WT or Orai1N223A expression plasmids or empty vector (Mock), as indicated. After 48 h cells were lysed and analyzed using Western blotting with anti-Orai1 antibody, followed by reprobing with anti-β-actin antibody for protein loading control. (**b**–**d**) MCF10A, MCF7-Orai1KO and MDA-MB-231-Orai1KO cells were transfected with Orai1WT or Orai1N223A expression plasmids as indicated. Forty-eight hours after transfection, fura-2-loaded cells were perfused with a Ca^2+^-free medium (100 µM EGTA added) and stimulated with TG (1 µM) followed by reintroduction of external Ca^2+^ (final concentration 1 mM) to initiate Ca^2+^ entry. Bar graphs represent TG-evoked Ca^2+^ release and entry in MCF10A (**b**), MCF7 (**c**) and MDA-MB-231 (**d**), expressed as fold increase experimental/control (Orai1WT-expressing cells). Data are mean ± SEM of 40 cells/day/3–5 days and were statistically analyzed using Student *t*-test. (**e**) Representative confocal images of Orai1WT and Orai1N223A cellular localization in MCF10A, MCF7 and MDA-MB-231 cells. The fluorescence signals of DAPI and PE were examined under a confocal laser scanning microscope. Scale bar: 25 µm.

**Figure 4 cancers-15-00203-f004:**
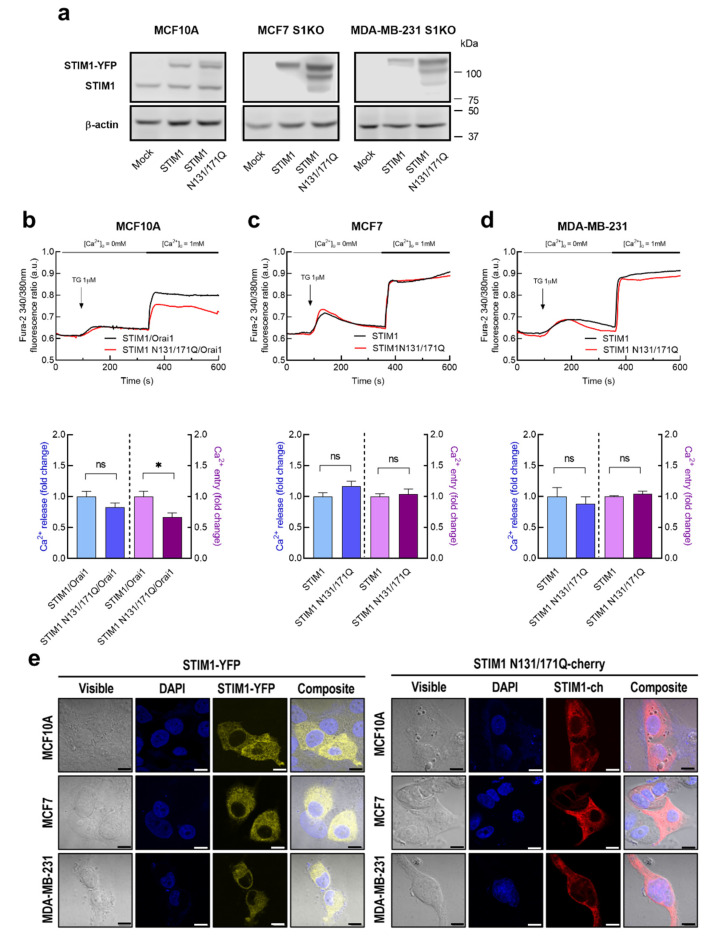
Characterization of Ca^2+^ mobilization in non-tumoral breast epithelial and breast cancer cells expressing STIM1WT and STIM1N131/171Q. (**a**) MCF7-STIM1KO (MCF7 S1KO) and MDA-MB-231-STIM1KO (MDA-MB-231 S1KO) cells were transfected with STIM1-YFP or STIM1N131/171Q-mCherry expression plasmids or empty vector (Mock), as indicated. MCF10A cells were transfected with STIM1-YFP or STIM1N131/171Q-mCherry in combination with Orai1. After 48 h cells were lysed and analyzed using Western blotting with anti-STIM1 antibody, followed by reprobing with anti-β-actin antibody for protein loading control. (**b**–**d**) MCF10A, MCF7-STIM1KO and MDA-MB-231-STIM1KO cells were transfected with STIM1-YFP or STIM1N131/171Q-mCherry expression plasmids as indicated. Forty-eight hours after transfection, fura-2-loaded cells were perfused with a Ca^2+^-free medium (100 µM EGTA was added) and stimulated with TG (1 µM; arrow) followed by reintroduction of external Ca^2+^ (1 mM). Bar graphs represent TG-stimulated Ca^2+^ release and entry in MCF10A (**b**), MCF7 (**c**) and MDA-MB-231 (**d**), expressed as fold change, experimental over control (STIM1-YFP-expressing cells). Data are mean ± SEM of 40 cells/day/3–5 days and were statistically analyzed using Student *t*-test (* *p* < 0.05). (**e**) Representative confocal images of STIM1-YFP and STIM1N131/171Q-mCherry cellular localization in MCF10A, MCF7 and MDA-MB-231 cells. The fluorescence signals of DAPI, YFP and mCherry were examined under a confocal laser scanning microscope. Scale bar: 25 µm.

**Figure 5 cancers-15-00203-f005:**
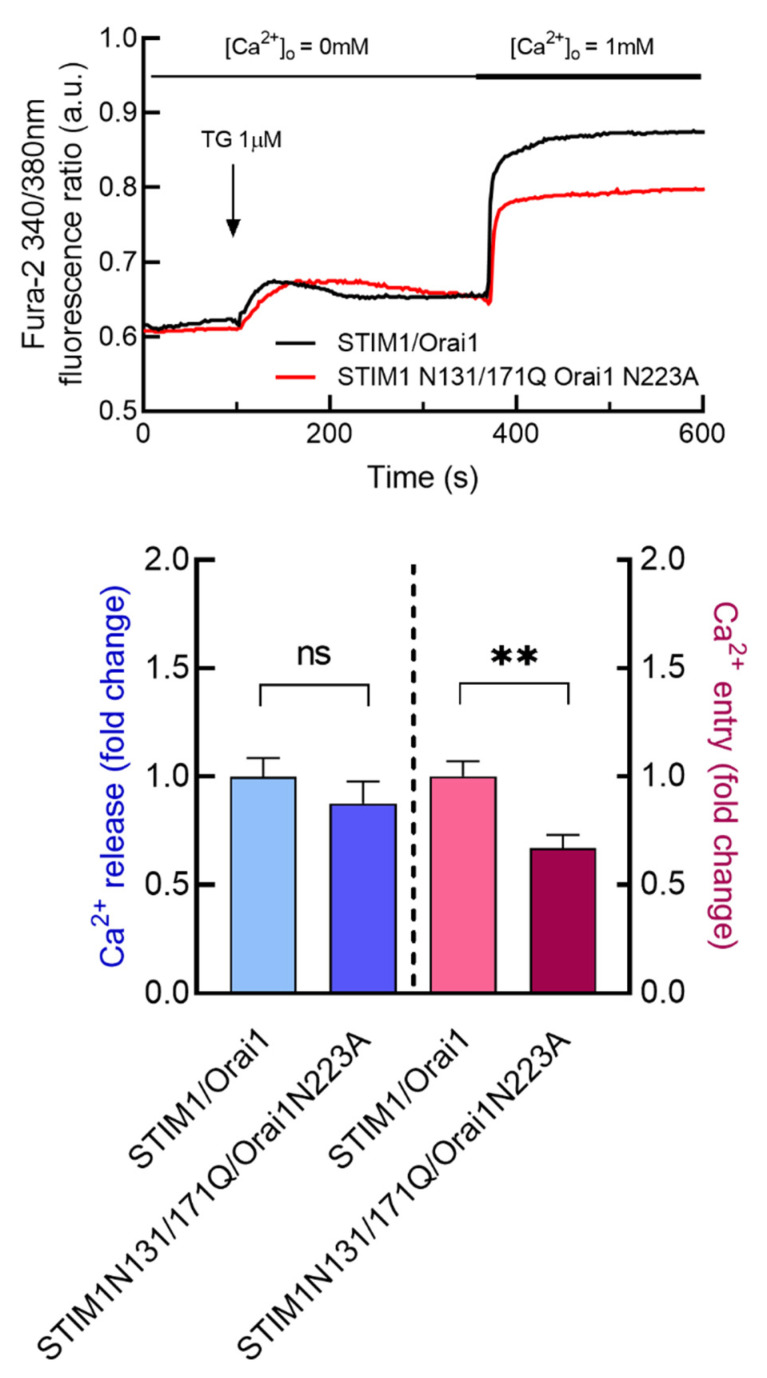
Expression of glycosylation-deficient mutants of Orai1 and STIM1 attenuate SOCE in non-tumoral breast epithelial cells. MCF10A cells were co-transfected with STIM1 and Orai1 or STIM1N131/171Q and Orai1N223A expression plasmids as indicated. Forty-eight hours after transfection, cells were loaded with fura-2. Fura-2-loaded cells were perfused with a Ca^2+^-free medium (100 µM EGTA was added) and stimulated with 1 µM TG followed by the addition of 1 mM Ca^2+^ to the extracellular medium. Bar graphs show TG-stimulated Ca^2+^ release and entry, expressed as fold change, experimental over control (STIM1/Orai1-expressing cells). Data are mean ± SEM of 40 cells/day/3–5 days and were statistically analyzed using Student *t*-test (** *p* <  0.01).

**Figure 6 cancers-15-00203-f006:**
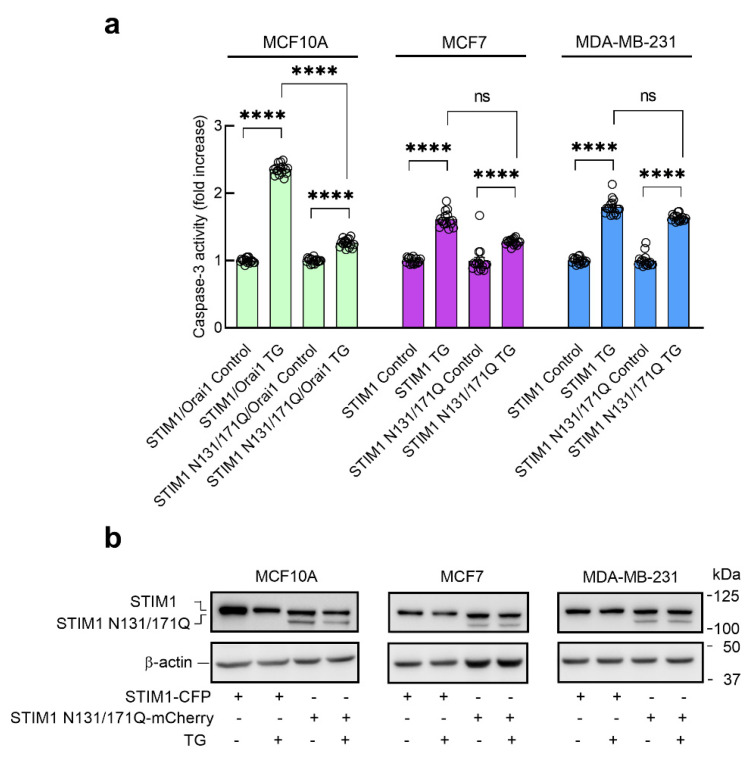
Effect of the expression of glycosylation-deficient STIM1 mutant on TG-evoked caspase-3 activation in non-tumoral breast epithelial cells and breast cancer cells. (**a**) MCF10A, MCF7-STIM1KO and MDA-MB-231-STIM1KO cells were transfected with STIM1-CFP or STIM1N131/171Q-mCherry expression plasmids, as indicated (MCF10A cells were co-transfected with Orai1). Forty-eight hours after transfection, cells were stimulated with TG (1 µM) in a medium containing 1 mM Ca^2+^ and caspase-3 activity was determined as described in Material and Methods. The scatter plots with bars represent TG-induced caspase-3 activity, expressed as fold increase experimental over control (unstimulated STIM1-YFP-expressing cells). Data are mean of 16 separate determinations and were statistically analyzed using Kruskal–Wallis test with multiple comparisons (Dunn’s test) to control cells (**** *p* <  0.0001). (**b**) MCF10A, MCF7-STIM1KO and MDA-MB-231-STIM1KO cells were transfected with STIM1-CFP or STIM1N131/171Q-mCherry expression plasmids, as indicated. Forty-eight hours after transfection, cells were stimulated with TG (1 µM) in a medium containing 1 mM Ca^2+^, lysed and analyzed by Western blotting using anti-STIM1 antibody, followed by reprobing with anti-β-actin antibody for protein loading control.

**Figure 7 cancers-15-00203-f007:**
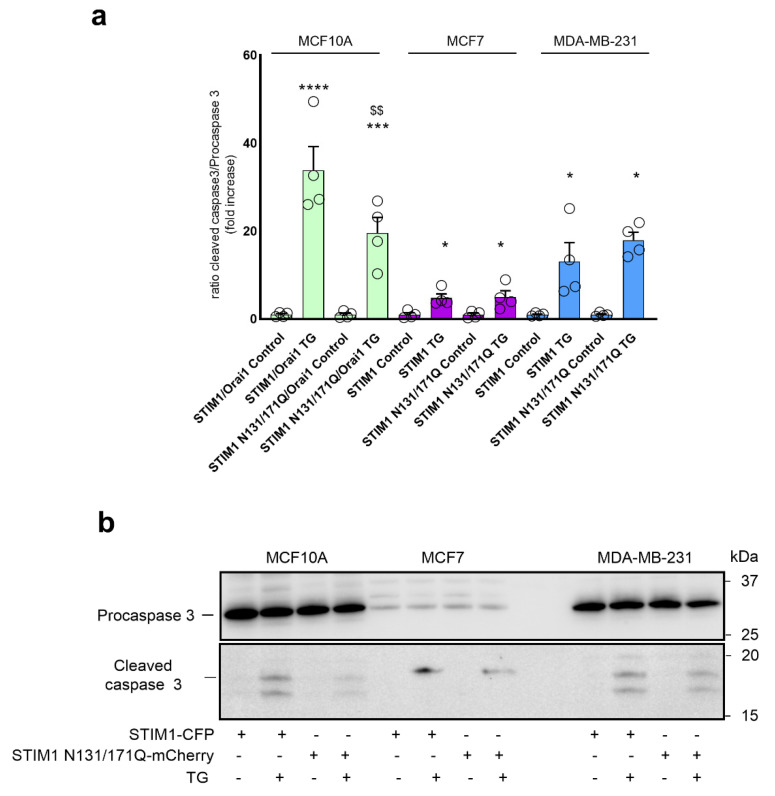
Effect of the expression of glycosylation-deficient STIM1 mutant on TG-evoked caspase-3 cleavage in non-tumoral breast epithelial cells and breast cancer cells. (**a**) MCF10A, MCF7-STIM1KO and MDA-MB-231-STIM1KO cells were transfected with STIM1-CFP or STIM1N131/171Q-mCherry expression plasmids, as indicated (MCF10A cells were co-transfected with Orai1). Forty-eight hours after transfection, cells were stimulated with TG (1 µM) in a medium containing 1 mM Ca^2+^ and caspase-3 cleavage was determined as described in Material and Methods. The scatter plots with bars represent TG-induced caspase-3 cleavage, expressed as the ratio of cleaved caspase-3/procaspase-3, and presented as fold increase experimental over control (unstimulated STIM1-YFP-expressing cells). Data are mean ± SEM of 4 separate determinations and were statistically analyzed using Kruskal–Wallis test with multiple comparisons (Dunn’s test) to control cells (**** *p* < 0.0001, *** *p* < 0.001 and * *p* < 0.05 as compared to control (unstimulated cells) and ^$$^
*p* < 0.01 as compared to cells expressing STIM1-CFP). (**b**) MCF10A, MCF7-STIM1KO and MDA-MB-231-STIM1KO cells were transfected with STIM1-CFP or STIM1N131/171Q-mCherry expression plasmids, as indicated. Forty-eight hours after transfection, cells were stimulated with TG (1 µM) in a medium containing 1 mM Ca^2+^, lysed and analyzed by Western blotting using anti-cleaved caspase-3 antibody (bottom panel) or anti-caspase-3 antibody (top panel).

## Data Availability

The data presented in this study are available on request from the corresponding author.

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
