# Peer review of "Store-Operated Calcium Entry in Breast Cancer Cells Is Insensitive to Orai1 and STIM1 N-Linked Glycosylation"

_cancers, 2022, doi:10.3390/cancers15010203_

Round 1
Reviewer 1 Report
Comments to the Author
The article investigated the role of Orail1 and STIM1 in SOCE activation, which is a preliminary study. The further mechanism failed to be further studied.
1) Sentence 195: The cell line should be MCF10A, and the description of BT20 should be added.
2) Sentence 206: “show a similar …… pattern”, however, in Fig.1, SKBR3 and HS578T don’t seem to show the same results?
3) Sentence 219: A “,” should be added before “we”.
4) Sentence 222: The reason to choose MCF7 and MDA-MB-231 for further study instead of other cell lines should be added.
5) Fig.6: More apoptosis related protein (e.g. the level of cleaved-caspase 3, BAX, Bcl2) should be examined.
Author Response
Dear Reviewer and Editor,
Thank you for your letter dated 13th November 2022 concerning the above manuscript. We would like to thank the Reviewer for his/her comments in order to improve the quality of the manuscript. We have modified the manuscript introducing all the requirements and suggestions of the Reviewer. Our responses to the points raised and the modifications made are detailed below.
Yours sincerely
Dr. Juan A. Rosado,
Responses to comments
Reviewer #1:
The article investigated the role of Orail1 and STIM1 in SOCE activation, which is a preliminary study. The further mechanism failed to be further studied.
1- Sentence 195: The cell line should be MCF10A, and the description of BT20 should be added.
Answer: We thank the Reviewer for drawing our attention to this point. We have corrected MCF10A and included the triple negative breast cancer cell line BT20.
2- Sentence 206: “show a similar …… pattern”, however, in Fig.1, SKBR3 and HS578T don’t seem to show the same results?
Answer: We thank the Reviewer for drawing our attention to this point. Indeed, there are different Orai1 glycosylation patterns in the cells investigated. Accordingly, we have rewritten the sentence that now reads: “Our results indicate that all the cells investigated show a similar STIM1 glycosylation pattern and cell type-specific Orai1 glycosylation patterns, as we found that in all the cells investigated, the pattern of Orai1 immunoreactivity appeared with distinct molecular masses (Figure 1).”.
3- Sentence 219: A “,” should be added before “we”.
Answer: we thank the Reviewer for his/her comment.
4- Sentence 222: The reason to choose MCF7 and MDA-MB-231 for further study instead of other cell lines should be added.
Answer: As requested, we have added the following sentence: “The MCF7 and MDA-MB-231 cell lines were chosen to perform the study as they are commonly used breast cancer cell lines for in vitro studies on ER+ and triple negative breast cancers [41]”. In this sentence, we cite a manuscript that reports that “the number of cell lines widely used for breast cancer studies is extremely small, with cell lines such as MCF7, T47D and MDAMB231 accounting for more than two-thirds of cell lines used in the associated studies”.
5- Fig.6: More apoptosis related protein (e.g. the level of cleaved-caspase 3, BAX, Bcl2) should be examined.
Answer: As requested, we have performed new experiments to test caspase-3 cleavage as an indicator of apoptosis. Our results, shown in Fig. 7, confirm the observations using the caspase-3 substrate Z-DEVD-AFC, and indicate that STIM1 glycosylation plays an important role in caspase-3 activation in non-tumoral cells but not in tumoral cells, thus suggesting that this mechanism might play a relevant role underlying apoptosis resistance in tumoral cells.
Reviewer 2 Report
In this article, the authors study the effect of treatment with tunycamicin of non-tumoral and cancer breast cells on SOCE (store-operated Ca2+ entry) proteins.
The title of the article is straightforward and apposit, the keywords are pertinent but provide no more info than those included in the title itself, so they do not improve the searchability of the article (therefore I suggest adding a couple that extend it).
The abstract is clear and appropriate in lengh and content.
The introdution is sufficient to provide the necessary background. The references are relevant, but I suggent to add some more recent ones, to show persistent interest in the field (less than 10% of the references cited in the introduction are to articles published in the last 3 years; the other references cited in the rest of the article are more recent (about 20%), but almost all of them refer to articles by the authors).
Materials and methods' description is clear and detailed.
Figures' readibility and labelling are good (included those of the supporting materials).
The discussion of data is lucid and the conclusions are consistent.
The scientific soundness and significance of content are solid, the originality and therefore the potential interest for the readers of the journal are average good, and so in my opinion this article is, all considered, well suited for publication (after suggested editings).
Author Response
Dear Dr. Liu,
Thank you for your letter dated 13th November 2022 concerning the above manuscript. We would like to thank the Reviewers for their comments in order to improve the quality of the manuscript. We have modified the manuscript introducing all the requirements and suggestions of the Reviewers. Our responses to the points raised and the modifications made are detailed below.
Yours sincerely
Dr. Juan A. Rosado,
Responses to comments
Reviewer #2:
In this article, the authors study the effect of treatment with tunycamicin of non-tumoral and cancer breast cells on SOCE (store-operated Ca2+ entry) proteins.
1-The title of the article is straightforward and apposite, the keywords are pertinent but provide no more info than those included in the title itself, so they do not improve the searchability of the article (therefore I suggest adding a couple that extend it).
Answer: We thank the Reviewer for drawing attention to this point. As suggested, we have included two more keywords to improve the searchability of the manuscript: posttranslational modifications and store-operated Ca2+ entry.
2-The abstract is clear and appropriate in lengh and content.
Answer: we thank the supporting comments.
3-The introdution is sufficient to provide the necessary background. The references are relevant, but I suggent to add some more recent ones, to show persistent interest in the field (less than 10% of the references cited in the introduction are to articles published in the last 3 years; the other references cited in the rest of the article are more recent (about 20%), but almost all of them refer to articles by the authors).
Answer: We thank the Reviewer for drawing our attention to this point. Accordingly, we have added five new references from 2022 in the introduction to show the current interest in this field.
4-Materials and methods' description is clear and detailed.
Answer: we thank the Reviewer for the supporting comments.
5-Figures' readibility and labelling are good (included those of the supporting materials).
Answer: we thank the Reviewer for his/her positive comments.
6-The discussion of data is lucid and the conclusions are consistent.
Answer: Again, we thank the Reviewer for his/her positive and supporting comments.
7-The scientific soundness and significance of content are solid, the originality and therefore the potential interest for the readers of the journal are average good, and so in my opinion this article is, all considered, well suited for publication (after suggested editings).
Answer: we appreciate the positive comments of the Reviewer.
Reviewer 3 Report
In this manuscript, authors investigated the role of STIM1 and Orai1 N-linked glycosylation in store-operated Ca2+ entry (SOCE) mechanism , in non-tumoral breast epithelial cells and in ER+ and triple negative breast cancer (TNBC) cells. They have shown that while STIM1 and Orai1 N-linked glycosylation has a significant effect on SOCE in non-tumoral breast epithelial cells, both ER+ 87 and triple negative breast cancer (TNBC) cells are unaffected by this mechanism. Their finding provides evidence for a mechanism to evade the regulation of Ca2+ influx in breast cancer cells.
A well organised manuscript with convincing figures. Just minor comments are:
1- Introduction: Among the mechanisms involved in the modulation of cell function by physiological agonist, store-operated Ca2+ entry (SOCE), regulated by the filling state of the intracellular Ca2+ stores, plays an essential role>> I don not know, something is not clear, please rephrase/
2- Sometimes sentences are very long, as example " In addition, a variety of regulatory proteins, including SARAF (SOCE-associated regulatory factor), which is involved in the mechanism underlying slow Ca2+-dependent inactivation of CRAC channels [10,11], 50 CRACR2A (CRAC regulator 2A), STIMATE, septins, golli and ORMDL3 (Orosomucoid 51 like 3) (for a review see [12]), as well as post-translational modifications, including phosphorylation at serine or tyrosine residues [13-16], S-glutathionylation [17], redox modulation and O- and N—linked glycosylation are involved in the modulation of CRAC channels (for a review see [18]).">> please split these to improve the readability for non native English speakers.
3- Please re-insert the magnification bars to the stained slides, use white bars or very dark, depending up on the background.
4- Under statistical analysis: P values <0.05 were considered statistically significant. why did not you mention the other values , P values <0.01, 0.001.
5- Discussion needs attention, the last 24 lines have no references. and the 1st two paragraphs include around 6 citations, So please rewrite parts of it and discuss more available data.
Author Response
Paper No: cancers-2019792
" Store-operated calcium entry in breast cancer cells is insensitive to Orai1 and STIM1 N-linked glycosylation as a mechanism to evade apoptosis "
Dear Reviewer and Editor,
Thank you for your letter dated 13th November 2022 concerning the above manuscript. We would like to thank the Reviewers for their comments in order to improve the quality of the manuscript. We have modified the manuscript introducing all the requirements and suggestions of the Reviewers. Our responses to the points raised and the modifications made are detailed below.
Yours sincerely
Dr. Juan A. Rosado,
Responses to comments
Reviewer #3:
In this manuscript, authors investigated the role of STIM1 and Orai1 N-linked glycosylation in store-operated Ca2+ entry (SOCE) mechanism , in non-tumoral breast epithelial cells and in ER+ and triple negative breast cancer (TNBC) cells. They have shown that while STIM1 and Orai1 N-linked glycosylation has a significant effect on SOCE in non-tumoral breast epithelial cells, both ER+ 87 and triple negative breast cancer (TNBC) cells are unaffected by this mechanism. Their finding provides evidence for a mechanism to evade the regulation of Ca2+ influx in breast cancer cells.
A well organised manuscript with convincing figures. Just minor comments are:
1- Introduction: Among the mechanisms involved in the modulation of cell function by physiological agonist, store-operated Ca2+ entry (SOCE), regulated by the filling state of the intracellular Ca2+ stores, plays an essential role>> I don not know, something is not clear, please rephrase/
Answer: We thank the Reviewer for his/her comments. We have modified this sentence, which now reads: “Among the mechanisms involved in the modulation of cell function by physiolog-ical agonist, store-operated Ca2+ entry (SOCE) plays an essential role. SOCE is a mechanism for Ca2+ influx regulated by the filling state of the intracellular Ca2+ stores.”
2- Sometimes sentences are very long, as example " In addition, a variety of regulatory proteins, including SARAF (SOCE-associated regulatory factor), which is involved in the mechanism underlying slow Ca2+-dependent inactivation of CRAC channels [10,11], 50 CRACR2A (CRAC regulator 2A), STIMATE, septins, golli and ORMDL3 (Orosomucoid 51 like 3) (for a review see [12]), as well as post-translational modifications, including phosphorylation at serine or tyrosine residues [13-16], S-glutathionylation [17], redox modulation and O- and N—linked glycosylation are involved in the modulation of CRAC channels (for a review see [18]).">> please split these to improve the readability for non native English speakers.
Answer: Again, we thank the Reviewer for drawing our attention to this point: We have modified this sentence that now reads: “In addition, a variety of regulatory proteins are involved in the modulation of CRAC channels. Regulatory proteins include SARAF (SOCE-associated regulatory factor), which is involved in the mechanism underlying slow Ca2+-dependent inactivation of CRAC channels [10,11], CRACR2A (CRAC regulator 2A), STIMATE, septins, golli and ORMDL3 (Orosomucoid like 3) (for a review see [12]) Similarly, a variety of post-translational modifications, including phosphorylation at serine or tyrosine residues [13-16], S-glutathionylation [17], redox modulation and O- and N—linked glycosylation, modulate CRAC channels (for a review see [18]).”.
3- Please re-insert the magnification bars to the stained slides, use white bars or very dark, depending up on the background.
Answer: As requested we have re-inserted clearer magnification bars in the images of Figures 3 and 4.
4- Under statistical analysis: P values <0.05 were considered statistically significant. why did not you mention the other values , P values <0.01, 0.001.
Answer. As we mention that P values <0.05 were considered statistically significant, this includes values smaller than 0.05, such as 0.01 or 0.001, which are also considered statistically significant.
5- Discussion needs attention, the last 24 lines have no references. and the 1st two paragraphs include around 6 citations, So please rewrite parts of it and discuss more available data.
Answer: As requested we have added some sentences discussing other data relevant to STIM1 and Orai1 N-linked glycosylation in SOCE and the development of distinct disorders.